# Improving Safety of Transportation of Dangerous Goods by Railway Transport

**Nijolė Batarlienė** 

Department of Logistics and Transport Management, Faculty of Transport Engineering,
Vilnius Gediminas Technical University, Plytines str. 27, LT-10105 Vilnius, Lithuania;
nijole.batarliene@vgtu.lt; Tel.: +370-5-2370-634

**Abstract:** The transport of dangerous goods by rail carries a high risk of accident and every effort should be made to ensure that such transport is carried out under the best possible safety conditions. The research objective was to analyze and identify the main risks associated with the transport of dangerous goods by rail as well as to identify and assess the main factors of safe transport in order to reduce the risk of accident. For this purpose, analysis of the literature, systematization, generalization, and evaluation by experts was applied. The article states that in order to ensure the safe transport of dangerous goods by rail, it is necessary to comply with the rules for loading and unloading dangerous goods, the established requirements and instructions, and technical conditions of wagons and their labelling as well as preventive measures to reduce the risk. Recommendations are provided on how to reduce accidents and incidents in the transport of dangerous goods by rail.

**Keywords:** dangerous goods; transportation; hazardous material; risk; safety; railway transport; accidents

## 1. Introduction

Thousands of tons of dangerous goods travel by all modes of transport every day. The transport of dangerous goods must comply with the relevant rules for the transport of such goods so that the goods can reach their destination safely. There is a risk of an event such as a spills, fire, explosion, chemical burns, or damage to the environment when transporting hazardous materials. Most goods are not considered as sufficiently dangerous to require special precautions during carriage. Some goods, however, have properties that mean that they are potentially dangerous if carried.

Due to the dangerous nature of goods, dangerous accidents in land transport often have dire consequences for the population and environment. The European Union is committed to achieving a 'zero accident' target by 2050, so it is important that safety programs include appropriate plans that take into account the safety requirements and development of the social economy.

Rail transport is superior to other modes of transport because it is universal, reliable, able to transport dangerous goods regardless of the time of year, climatic conditions, and transport options. Dangerous goods can be transported by rail in large quantities and over long distances.

Special safety requirements are required for the transport of dangerous goods by rail both internationally and nationally. It is necessary to regulate, control, and inspect the transport of dangerous goods by rail because of their characteristics and the real dangers. International requirements (RID) apply to the transport of dangerous goods, packaging, and marking of wagons [1]. Other requirements for carriers such as the storage of dangerous goods and other logistic functions are also important.

The transport of dangerous goods by rail presents a considerable risk of accident. Measures should therefore be taken to ensure that such transport is carried out under the best possible conditions of safety [2]. During the transport of dangerous goods, there may be a risk of an accident due to the absence of incorrectly chosen packaging materials or markings, and the fault of other road users or

climatic conditions. In the transportation of dangerous goods, it is impossible to avoid risk; however, it is possible to manage and reduce the risk increasing factors to a minimum.

Having analyzed accidents in the transportation of dangerous goods, it can be seen that accidents or incidents in the transportation of dangerous goods cause more problems than in the transportation of usual goods [3]. The problems of the transportation of dangerous goods are important not only to enterprises engaged in the transportation of dangerous goods, but also to all of the institutions responsible for the control of dangerous goods. In the transportation of dangerous goods, risk and possible danger to the safety of the public and the environment are inevitable [4–6].

The transport of dangerous goods by railway poses an imminent risk of accident, which can injure people, their property, and the environment [7–9]. Railway safety indicators are influenced by several factors such as the risk factors of people working on the section of the railway as well as infrastructure risk factors. Infrastructure includes the basic facilities and systems that serve a country's territory including the services and facilities necessary for its economy to function. The scope of the journal Infrastructure covers all modes of transport including rail transport infrastructure, and includes railway engineering and sustainable infrastructure.

The aim of the article was to analyze and identify the main risks associated with the transport of dangerous goods by rail as well as to identify and assess the main factors of safe transport, in order to reduce the risk of accidents and provide recommendations on how to transport dangerous goods safely.

Working methods include the analysis of the literature, systematization, generalization, and evaluation by experts.

The rest of this paper is organized as follows. Section 2 provides a literature review to specify the motivation and background of the related studies. Articles are discussed that help to find the specifics of the transport of dangerous goods by rail, and the importance and relevance of the methods they use. Section 3 presents an analysis of the problem and provides information and statistics on the transport of dangerous goods and accidents. Section 4 presents the research methodology, and Section 5 discusses the results of this study. Section 6 provides the limitations and recommendations. Accident reduction measures for the transport of dangerous goods by rail are also provided in this section. Discussion and conclusions are discussed in Section 7.

## 2. Literature Review

In the development of transportation technologies, the solution of theoretical problems and the use of practical tools play an important role, while improving the process of the safe transportation of dangerous goods.

The main document that regulates the transport of dangerous goods by rail is the Regulation concerning the International Carriage of Dangerous Goods by Rail (RID), which is adjusted and supplemented every two years. RID applies to the international carriage of dangerous goods by rail between the 44 existing RID Contracting States in Europe, Asia, and North Africa. In Member States of the European Union, RID also applies to national as well as international transport. RID is harmonized with the United Nations' Recommendations on the Transport of Dangerous Goods, which serve as the basis for all the modal dangerous goods regulations. There is also close coordination with the dangerous goods regulations for road (ADR) and inland waterways (ADN).

The classes of dangerous goods (DG) according to RID are as follows:

Class 1: Explosive substances and articles
Class 2: Gases
Class 3: Flammable liquids
Class 4.1: Flammable solids, self-reactive substances, polymerizing substances and solid desensitized explosives
Class 4.2: Substances liable to spontaneous combustion
Class 4.3: Substances which, in contact with water, emit flammable gases
Class 5.1: Oxidizing substances

Class 5.2: Organic peroxides

Class 6.1: Toxic substances

Class 6.2: Infectious substances

Class 7: Radioactive material

Class 8: Corrosive substances

Class 9: Miscellaneous dangerous substances and articles [1].

As the transport of dangerous goods increases, so does the risk they pose during transport. The literature states that it is necessary to guarantee the safety of transportation processes, given that dangerous substances pose a significant threat not only to the environment and people, but also to the entire transport infrastructure. Diernhofer et al. [10] stated that risks are often not due to the properties of hazardous substances, but to human error in production and transport.

Railway safety is vital to the economic and social development of each country. However, studies to assess the safety of the country's rail transport in transporting dangerous goods are still at an early stage. In recent years, some progress has been made in theoretical research and practice to provide a better understanding of the measurements of railway safety performance.

One important aspect that needs to be considered is the risk of population exposure. The mathematical models are tested in the case of an LPG company in Korea [11]. The study aimed to provide a systematic approach for optimizing a fuel logistics network (FLN) accounting for not only transportation costs, but also the process related risks and the relevant externalities. The main contribution of the study is to highlight the non-financial operational factors, including safety, emissions, and congestion, which have a direct impact on the sustainability of the fuel sector.

Several studies using certain accident probability factors have been discussed in the literature. For example, according to Chakrabari and Patrikh [12], the risk can be calculated according to the probability of the event and the consequences of the incidents. The authors state that the risk of transporting dangerous goods depends on three main factors, namely the intensity of daily traffic, the number of accidents, and the length of the route.

It can be found in the literature that the transport of dangerous goods is associated with the risk of an incident due to the fault of other road users, climatic conditions, improper choice of packaging materials, or lack of labelling. Although the transport of dangerous goods is risky, there is a need to learn to manage the risk and thus reduce it [13–15].

According to Fabiano et al. [16], the effectiveness of emergency planning can usually be assessed in terms of system speed and reliability. They argue that the safety and efficiency of transport systems must be considered as a strategic goal. However, the article put more emphasis on road transport.

When transporting dangerous goods, every transport process is very important and there is a risk of an incident throughout the logistics chain. Kršák et al. [17] argue that it is necessary to study the factors that influence transport risk in order to ensure safety. For loading, packing, marking, storing, preparing the necessary documents for transportation, and performing other operations for dangerous goods, highly qualified personnel are required to ensure that everything is done without mistakes, and in this way, incident risk could be avoided.

Ghazinoory and Kheirkhah [18] agree that hazardous materials are continuously moved between all countries. These movements are naturally dangerous as the release of hazardous substances as a result of an accident can lead to deaths and irreparable damage to the environment.

Researchers have emphasized that special attention needs to be paid to the transport of dangerous goods not only during the preparation and loading of such cargo, but also during transportation until the cargo reaches its destination [19–24]. Blanco [25] argues that loading and unloading dangerous goods is less dangerous than transportation by road, but also covered some of the potential risks posed by these operations. However, the author did not examine more factors related to the safety of the transport of dangerous goods. In addition, it only analyzed the road safety aspects.

Conca et al. [26] stated that the safety of the transport of dangerous goods was a very important task in the transport planning process. Jarašūnienė and Jakubauskas [27] proposed improving road

safety with passive and active safety systems for intelligent vehicles, but was not particularly suitable for the transport of dangerous goods by rail.

Najib et al. [28] suggested the use of information systems for the transport of dangerous goods as it is possible to quickly obtain the necessary information regarding the location of the vehicle, the condition of the load, or any other necessary information at the right time. Scientific papers related to the risk assessment of the transport of dangerous goods have been submitted by many authors including Milazzo [29], Grzegorczyk [30], Fabiano [15], Xuan et al. [4], Alyami et al. [31], and others.

Zhao et al. [32] found that the three most influential factors in transportation accidents of hazardous materials were human factors, the transport vehicle and facilities, and the packing and loading of hazardous materials in China, but these factors were more related to road transport.

Bouissou et al. [33] described the QRA system used in France. With the help of this system, it is possible to assess the risks during the transport of dangerous goods as well as assess the consequences of events and possible probabilities of incidents.

Tomasoni et al. [34] described the importance of warning signs in the transport of dangerous goods. These authors pointed out that the labeling of dangerous goods containers and tanks with warning signs was the most important aspect for providing information about a specific dangerous substance.

The transport of dangerous materials is important at every stage of the supply chain including the first and last mile delivery. Slabinac [35] presented innovative transport technologies in "last-mile" delivery in developed European countries that provided ecological and social sustainability as well as an increased competitiveness of the suppliers. "Last-mile" delivery in an urban environment represents a constituent part of city logistics as the final leg conducted over short distances to reach the customer. According to Macioszek [36], in freight transport, the first and the last mile are most cost intensive. Schliwa et al. [37] argued that local authorities play a key role in creating conditions that incentivize large logistic companies to integrate cargo cycles into their supply chain and hence drive a long-term modal shift.

Summarizing the analysis of the literature, it can be stated that it is very important to use all theoretical and practical knowledge to make the transportation of dangerous goods safer. However, analytical problem solving is complicated by the cost and duration of the formalization of dangerous goods transportation processes and experimental research. The articles have mainly analyzed the requirements for land transport in general, but do not single out the main risks and the main factors that determine the likelihood of dangerous goods accidents by rail, nor do they provide recommendations on how to ensure the safety of dangerous goods by rail.

## 3. Problem Analysis

The volume of dangerous goods transportation is increasing every year in EU countries. As a result, not only are corporate transport fleets expanding, but they are also leading to higher accident risks [38].

According to Lithuanian statistics, Figure 1 shows the most common quantities of dangerous goods carried by railways in Lithuania. We can see that flammable liquids of hazard class 3 are the most commonly carried substances.

The most common substances in this hazard class are diesel, varnish, and petrol. Between 2013 and 2017, shipments of these materials were quite stable and amounted to about 10,000 thousands of tons.

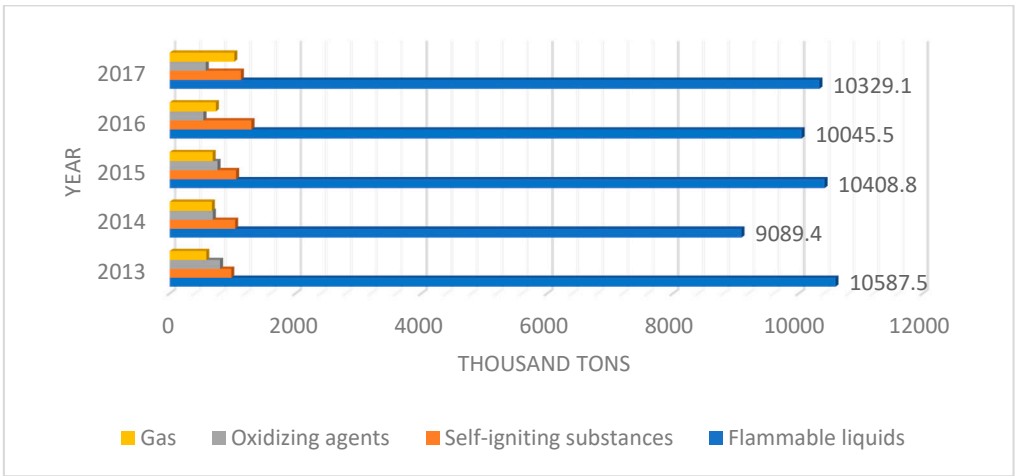

**Figure 1.** Quantities of dangerous goods transported by railway according to hazard classes from 2013 to 2017 (thousand tons) in Lithuania [39].

The second group, based on the amount of material most frequently transported, includes self-igniting materials such as activated carbon, phosphor, and so on. No significant fluctuations in this class of transport were observed, reaching from 1000 thousand tons up to 1300 thousand tons per year. The third group is oxidizing substances, the amount of which transported by 2015 on average was about 750 thousand tons per year. Starting with 2015, the volume of shipments of these materials began to decrease and in 2017, the amount reached 572,6 thousand tons per year. The last group, according to the frequency of transportation, belongs to Class 2, and include gases such as aerosols, propane, or butane. Carriage volumes of this class have increased since 2015 and in 2017 reached 1026 thousand tons.

Incidents in railway transport can be divided into three groups:

- Derailment is an incident that due to the collision of trains and rolling-stock or train and railway vehicle derailment, collision of trains and rolling-stock with road transport or other means of transport, one or more persons lose their lives, or five or more persons are injured, or damage amounting to at least 2 million EUR is done to railway infrastructure, rolling-stock, the environment, or property of legal persons as well as any other similar incident due to which it is impossible to control and (or) manage traffic safety in railways.
- Accident is an incident of railway transport when trains, rolling-stock collide, trains collide with rolling-stock or buildings, installations, rolling-stock derails, and railway transport accidents occur at crossings due to moving rolling-stock where more than four persons are injured and fire breaks out in railway transport.
- Major accident is an accident when there is at least one moving rolling-stock and at least one person has lost their life or was gravely injured or great damage was done to rolling-stock, railing, other installations, or environment, or traffic was cancelled for a long time. Significant damage is estimated at € 150,000 or more. This definition does not include accidents in workshops, storehouses and depots.

Incidents and accidents occur quite frequently in EU Member States, but it is difficult to find statistics on the number and type of incidents involving the transport of dangerous goods in either the Eurostat database or the Statistics Lithuania database. Although an analysis of the data provided shows that the number of accidents is declining, it still remains high.

Figure 2 shows the number of accidents involving dangerous goods by rail in the European Union, according to Eurostat [40].

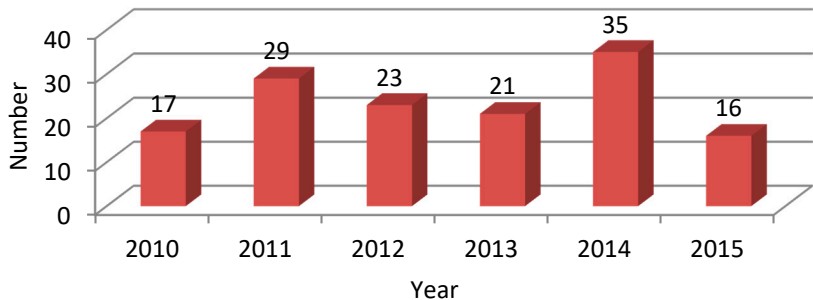

**Figure 2.** Number of accidents involving the transportation of dangerous goods by railway transport in the European Union from 2010 to 2015 [40].

Figure 2 shows that the majority of accidents involving the transport of dangerous goods by rail were in 2011 and 2014, an increase from 21 to 35 accidents per year. In drawing conclusions, it should be noted that these figures do not include accidents in which no accident occurred.

The analysis of dangerous goods transportation by railway describes the needs, organizational, and technical measures to achieve a safe and scientifically based dangerous goods transportation process. To ensure transport safety, the factors influencing transport hazards must first be investigated.

## 4. Research Methodology

Expert evaluation was used to conduct the study. This is a procedure for reconciling the opinions of individual experts and forming a common solution. We termed an expert as someone who has experience and knowledge [41]. The reliability of an expert evaluation depends on the size of the expert group, the composition of the experts according to their specialty, and the characteristics of the experts.

Expert evaluation methods are applied:

- where the information is large, but qualitative or multi-criteria; and
- when information is inadequate, and it is in prediction tasks [42].

The expert review methodology is based on the following assumptions:

- that the expert has a wealth of rationally processed information and can therefore be a source of information; and
- that the expert group's opinion differs little from the actual solution to the problem.

Expert evaluation is conducted in accordance with the ethical requirements of social research:

- respondents were chosen to have sufficient competence to answer the questions;
- the consent of respondents to participate in the survey; and
- the researcher has no influence on the respondents [43].

The survey conducted in the autumn of 2019 involved 47 experts. It was very important to find suitable experts for the research in all of the analyzed companies. The determination of the acceptable number of experts was based on the methodological assumptions formulated in classical test theory, which states that the reliability of aggregate decisions and the number of decision-makers (in this case, experts) is linked by a rapidly extinguishing nonlinear relationship. The questionnaires were distributed to 65 employees of JSC "Lietuvos Geležinkeliai" ("Lithuanian Railways"), who worked directly with the transportation of dangerous goods. The survey was conducted by sending questionnaires by email or in meetings with railway staff. Forty-seven questionnaires were received.

During a qualitative study in a railway company, experts with a length of service ranging from five to 15 years working with dangerous goods were interviewed. The survey revealed which problems

were most often encountered in organizing the transport of dangerous goods, what information technologies were used in transport, how many traffic accidents occurred in the last three years, and what the reasons were for their increase as well as in what additional ways JSC "Lietuvos Geležinkeliai" trains its drivers in the transport of dangerous goods [44].

In order to find out how often dangerous goods were transported by the company during the year, what problems were encountered, and which factors most often led to traffic accidents or accidents, a questionnaire was developed. The following tasks were set for the research:

- to find out how often rail transport company "Lietuvos Geležinkeliai" faced the organization of dangerous goods during the year;
- what were the most common classes of dangerous goods carried by railway transport;
- to identify the most common problems encountered in organizing the transport of dangerous goods by rail;
- to find out whether specially adapted information systems were used for the transport of dangerous goods;
- to determine whether accidents with dangerous goods had occurred in the railway in the last year as well as the common causes of their accidents;
- to find out if additional training is provided for train drivers to transport dangerous goods; and
- to find out which factors of the transport of dangerous goods by rail had the greatest impact on the accident.

Questionnaire data were processed and analyzed using Microsoft Excel. The results of the research are presented in the next section.

## 5. Research Results

After receiving the results of the expert surveys, the information was processed. Comparative analysis of research data was used. For the purpose of comparability, the values were expressed as a percentage. Figure 3 shows that the cargo was mainly transported in four hazard classes, namely Class 2: gases, Class 3: flammable liquids, Class 8: corrosive substances, and Class 9: miscellaneous dangerous substances.

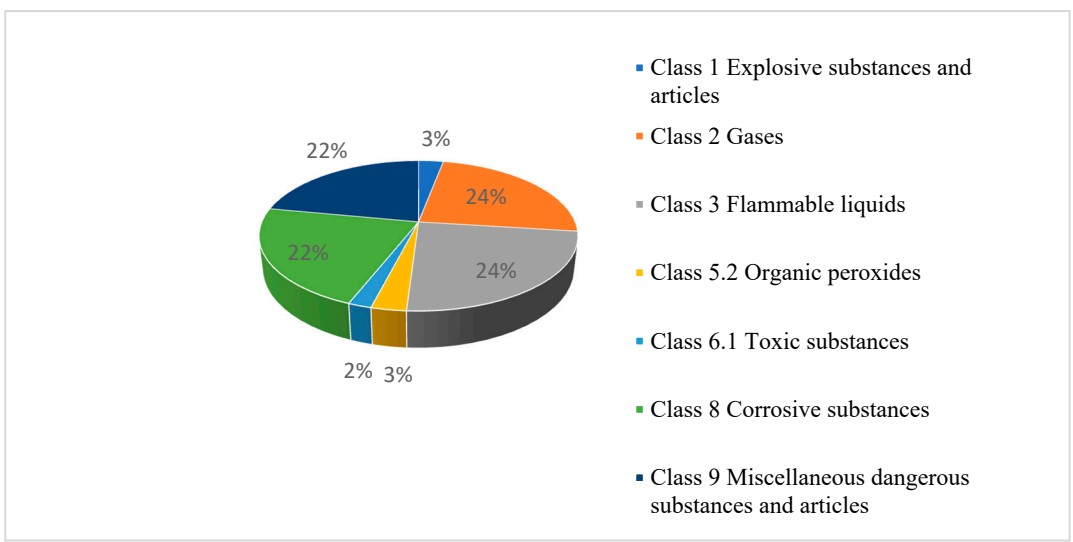

**Figure 3.** Transport of dangerous goods by rail in Lithuania according to hazard classes in 2019.

JSC "Lietuvos Geležinkeliai", when organizing the transport of dangerous goods, usually faces the following problems:

- incorrectly completed consignment note;

- non-compliance with the rules for gluing orange warning signs;
- improperly marked cargo;
- failure to provide detailed cargo information;
- detachment of warning signs from cystic wagons due to bad weather conditions; and
- defects of tank wagons (leaks, packaging defects) during loading.

When interviewing experts about the information technologies used in the company that facilitate the planning and organization of dangerous goods, three information systems that are currently in use were presented:

- IS "Krovinys" ("Cargo");
- "OPKIS"; and
- "E-krovinys" ("E-cargo").

These three information systems are used in organizing the transport of dangerous goods by rail in Lithuania. Experts were asked about the purchase of additional information systems. The answer was that no new systems were needed yet, and that existing ones were enough.

However, not always based on any information system, the number of accidents or incidents was reduced. A survey of experts about accidents that had occurred in the last three years led to the conclusion that the company had an average of three to four accidents per year.

Five factors were provided for the survey, which had to be evaluated in a 5-point system based on their importance (very important—5, important—4, important on the average—3, unimportant—2, and completely unimportant—1). The summarized experts' answers, their evaluation averages, and standard deviations of the answers are provided in Table 1. The averages of the data and correlation of factors were calculated.

**Table 1.** Evaluations factors by the experts.

| Evaluation/ Factor | Scoring System | | | | | Evaluation Average | Standard Deviation |
|---|---|---|---|---|---|---|---|
| | 5 Points | 4 Points | 3 Points | 2 Points | 1 Point | | |
| Compliance of wagons, containers and tanks with RID requirements | 31 | 27 | 58 | 41 | 26 | 2.98 | 1.28 |
| Risk of fire and explosion | 39 | 57 | 44 | 25 | 18 | 3.40 | 1.24 |
| Exposure to dangerous substances during loading and unloading | 65 | 53 | 33 | 25 | 7 | 3.79 | 1.18 |
| Qualification of train driver | 25 | 29 | 38 | 37 | 54 | 2.64 | 1.40 |
| Technical requirements for the safety of internal and external valves | 37 | 28 | 84 | 26 | 11 | 3.33 | 1.09 |

The highest risks were identified by experts as being due to the loading and unloading properties of hazardous substances (average rating 3.79), fire and explosion risks (3.40), and safety requirements for internal and external valves (3.33). These three factors should be given the most attention. While the evaluation of the compliance of wagons, containers, and tanks with RID requirements and the qualification of the train driver showed averages of less than three points, in order to ensure the proper security of dangerous goods, these two factors must also be evaluated.

After analyzing the answers of the experts, it can be added that other circumstances (proper planning, training, safe and proper packaging, etc.) should be considered in order to reduce the accidents and losses of JSC "Lietuvos geležinkeliai".

During the investigation, it was found out that JSC "Lietuvos geležinkeliai" provided driver training in accordance with the requirements of RID. Each driver's cab must have written instructions

in accordance with the RID. They were also trained in the notification procedure in the event of an incident and the action to be taken, and great effort was made to carry out all driver training within the company.

## 6. Practical Implications

The variety of data categories associated with this case study presented several limitations that could be improved in future work, starting with incident and accident data that did not have a description of the scene, that would allow an assessment of the probabilities of the event. More accurate data could be obtained, and an information system installed. The probability of an incident could also be improved by developing time of day models. Another area for improvement would be data on a comprehensive study of the health costs associated with exposure to a hazardous substance.

European safety and health policy concern the calculation and assessment of risks and the application of preventive measures. Priority is given to eliminating the risk at the source. These principles must be applied in the development of a railway safety plan, which includes operational planning, staffing, and measures. It is recommended that accidents and incidents are registered and analyzed. This analysis could then serve as a solution to the decisions that need to be made for the safe transport of dangerous goods.

In order to reduce the risk of accidents, general conditions must be laid down to ensure compliance with the requirements for the transport of dangerous goods by rail. The first is to ensure compliance with the requirements for the identification of dangerous substances. Compliance with the legal provisions is very important, considering the special requirements for the transport of dangerous goods. The specific requirements for the transport of dangerous goods need to be further assessed, and ensuring staff training is also a very important factor.

It is very important to ensure the control and constant inspection of various equipment such as the inspection of equipment for the transport, loading, and unloading of dangerous goods. Implementation of controls is also needed to ensure compliance with loading and unloading rules. It must always be ensured that personnel handling or loading and unloading dangerous goods follow work procedures and instructions. Controls are essential to ensure that vehicles have documentation and safety equipment with regard to their compliance.

An important task is to implement measures to prevent any accidents, incidents, or serious breaches. Once any accident or incident involving the transport of dangerous goods has taken place, safe emergency procedures must be applied and, finally, investigations must be carried out into the causes of major accidents, incidents, and damage during transport, and the loading or unloading of dangerous goods.

In order to ensure a safe process for the transportation of dangerous goods by rail, different controlling institutions that would be responsible for the safe transportation of dangerous goods should be combined.

In summary, measures to reduce the risk of transporting dangerous goods by rail can include maintaining the technical condition of rail tanks and containers; quality assurance of loading, transport, and unloading of dangerous goods; recruitment of highly qualified locomotive drivers; compliance with established requirements and instructions; ensuring the training, experience, and knowledge of the train driver and other actors involved in the transport of dangerous goods; tracking of train drivers during the transport of dangerous goods; use of information systems for the transport of dangerous goods; implementation of preventive measures, during which attention would be paid to the technical condition of train wagons, ensuring the tightness of containers and tanks, and their marking as well as the correct completion of documents and the provision of instructions to locomotive drivers. The results of the study showed that investment is needed in rail transport tanks that are specially prepared for the transport of dangerous goods.

Preventive action is needed on railways and level crossings to encourage drivers and pedestrians to comply with road traffic rules as well as to take all measures to avoid similar situations, given the accidents that have already occurred.

These findings provide an empirically supported theoretical basis for dangerous goods transportation enterprises to take corrective and preventative measures to reduce the risk of accidents.

## 7. Conclusions

Transportation of dangerous goods is one of the most complex and safety requiring transportation technologies. Due to the peculiarities and risk, their transportation must be precisely controlled, regulated, and handled.

An analysis of rail freight and the likelihood of accidents leads to the conclusion that, although the number of all accidents is declining, accidents involving dangerous goods occur quite frequently. Major accidents were avoided during the year under investigation, but there is no guarantee that severe consequences will be avoided in the future.

Based on the data of the research, losses due to accidents and losses due to other factors were distributed in a very similar way. Enterprises should pay equal attention to the mounting of goods (prevention of losses incurred during accidents) and measures that help to prevent accidents as well as measures contributing to the reduction of accident consequences.

The losses incurred could be reduced just by the complex use of different measures, as individual measures only reduce the probability of specific risk occurrence, while the sum of adequately chosen measures allows for the reduction of risk level to a tolerable limit.

Having conducted the research, it was established that it is necessary to apply prevention measures and inform all traffic participants about the danger of transporting dangerous goods, therefore personnel who are aware of this would try to maintain a safer distance and adequately assess the risk of danger. This could reduce the number of accidents.

Accidents in the transportation of dangerous goods occur not only due to the fault of the locomotive driver, but also takes place due to the technical reasons or the fault of other road users. The most important factors were identified by a qualitative expert survey, which identified five factors that have the greatest impact on the occurrence of accidents.

According to the results of the questionnaire, it was found that the greatest risks were associated with the effects of hazardous substances on the environment and humans, particularly during loading and unloading, risk of fire and explosion as well as failure to ensure the safety of the internal and external safety valve. These three factors should be given the most attention.

The number of rail accidents can be reduced if the quality of loading, transport, and unloading is ensured; it is also essential to ensure the preparation, experience, and knowledge of the train driver and other actors involved in the transport of dangerous goods. It is necessary to implement preventive measures, which would pay attention to the technical condition of trains, ensuring the tightness of containers and tanks, and their markings. No less important aspect is the installation of new technologies, etc.

An important task is to assess the risk. This can be done in different ways and by using different models and methods, depending on certain risk factors and the results to be achieved.

Further work on the risk of transporting dangerous goods by rail may focus on a risk assessment system based on a specific location and time of day. In this way, a theoretical approach to emergency planning and optimization could be developed. The first step would be to create a database useful for realistically estimating the frequency of incidents and accidents on a given route using multidimensional statistical analysis.

**Funding:** This research received no external funding.

**Acknowledgments:** The author would like to thank the anonymous experts and reviewers for their valuable suggestions.

**Conflicts of Interest:** The author declares no conflicts of interest.

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
