# Peer review of "Improving Safety of Transportation of Dangerous Goods by Railway Transport"

_infrastructures, doi:10.3390/infrastructures5070054_

Round 1

Reviewer 1 Report

Abstract: you do not need to point out the different sections of the abstract; for example, "Working methods: analysis of the scientific literature, systematization, generalization, evaluation by experts." can be replaced by "For this purpose, analysis of the scientific literature, systematization, generalization, evaluation by experts is applied." 

"The results of the survey are presented and how the experts assess the main factors related to the risk of railway transport." is unnecessary. 

Introduction: this section has to be improved. The readability of the introduction is particularly poor. The first paragraph is quite vague. There is no logical connection between the different paragraphs of the introduction part. Start with a background and the role of transportation (all modes). Then narrow it down to railway transportation. Besides, you need to make it clear how your work is important to the "Infrastructure" readership.

The academic gap is not clear. You need to state the gap based on a review of the most relevant (similar) works in the literature.

Literature review: this part should, as well, follow a logical order. Please consider 2-3 sub-section, and finish the section by a paragraph summarizing the review of literature, and re-stating the scientific gap your research is going to address.

The research methodology section needs major improvement. Once again, you should assume that some of your readers are students, and would like to replicate your method in their study.

Research results: there should be a connection between the research objectives (research question) you state in the first paragraph, and the results of your study. This connection is rather weak in your manuscript.

Please rename "limitation and recommendation" to "practical implications". You need to make sure the points mentioned in this part are well supported by the results in the previous section.

In the last two sections, you should not use bullet points and numbering. I could spot some minor plagiarism in some sections of your work. Please make sure you are paraphrasing the sentences if they are taken from other studies.

Please make sure a decent language-proof reading service is used before your next submission.

Author Response

Response to Reviewer 1 Comments

Thank you very much for your valuable comments on the manuscript (843293) with the title “Improving Safety of Transportation of Dangerous Goods by Railway Transport”. I have tried my best to revise and improve the manuscript and made many changes in the manuscript according to your helpful comments.

Point 1. Abstract: you do not need to point out the different sections of the abstract; for example, "Working methods: analysis of the scientific literature, systematization, generalization, evaluation by experts." can be replaced by "For this purpose, analysis of the scientific literature, systematization, generalization, evaluation by experts is applied."

Response: Thank you for this constructive suggestion. The abstract has been restructured.

Point 2. "The results of the survey are presented and how the experts assess the main factors related to the risk of railway transport." is unnecessary.

Response: Thank you for this helpful comment. This sentence has now been deleted in the Abstract.  

Point 3.  Introduction: this section has to be improved. The readability of the introduction is particularly poor. The first paragraph is quite vague. There is no logical connection between the different paragraphs of the introduction part. Start with a background and the role of transportation (all modes). Then narrow it down to railway transportation. Besides, you need to make it clear how your work is important to the "Infrastructure" readership.

Response: Thank you for this constructive suggestion. The introduction has now been amended and supplemented. The first 3 paragraphs (lines 23-36) now look like this:

„Thousands of tons of dangerous goods travel by all modes of transport every day. The transport of dangerous goods must comply with the relevant rules for the transport of such goods so that the goods can reach their destination safely. There is a risk of an event, such as a spill, such as fire, explosion, chemical burns or damage to the environment when transporting hazardous materials. Most goods are not considered sufficiently dangerous to require special precautions during carriage. Some goods, however, have properties which mean they are potentially dangerous if carried.

Due to the dangerous nature of goods, dangerous accidents in land transport often have dire consequences for the population and the environment. The European Union is committed to achieving the 'zero accident' target by 2050, so it is important that safety programs include appropriate plans that take into account safety requirements and the development of the social economy.

Rail transport is superior to other modes of transport because it is universal, reliable, able to transport dangerous goods regardless of the time of year, climatic conditions and transport options. Dangerous goods can be transported by rail in large quantities and over long distances.“

Point 4.  The academic gap is not clear. You need to state the gap based on a review of the most relevant (similar) works in the literature.

Response: Thank you for this constructive suggestion. The academic gap is now highlighted (lines 107-110):

“Railway safety is vital to the economic and social development of each country. However, studies to assess the safety of the country’s rail transport in transporting dangerous goods are still at an early stage. In recent years, some progress has been made in theoretical research and practice to provide a better understanding of measurements of railway safety performance.”

Point 5. Literature review: this part should, as well, follow a logical order. Please consider 2-3 sub-section, and finish the section by a paragraph summarizing the review of literature, and re-stating the scientific gap your research is going to address.

Response: Thank you for this comment. It was done. The literature was reviewed carefully again, and this section has been improved. I could not change the 2-3 sub-sections because one reviewer insisted on the inclusion of such material.

The main additions were made as follows (lines 171-177):

Summarizing the analysis of the literature, it can be stated that it is very important to use all theoretical and practical knowledge to make the transportation of dangerous goods safer. However, analytical problem solving is complicated by the cost and duration of formalization of dangerous goods transportation processes and experimental research. The articles mainly analyze the requirements for land transport in general, but do not single out the main risks and the main factors that determine the likelihood of dangerous goods accidents by rail, nor do they provide recommendations on how to ensure the safety of dangerous goods by rail.”

Point 6. The research methodology section needs major improvement. Once again, you should assume that some of your readers are students, and would like to replicate your method in their study.

Response: Thank you for this comment. It was done. The main additions were made as follows (lines 264-276):

“The following tasks were set for the research:

  • To find out how often rail transport company “Lietuvos Geležinkeliai” face the organization of dangerous goods during the year.
  • What are the most common classes of dangerous goods carried by railway transport?
  • To identify the most common problems encountered in organizing the transport of dangerous goods by rail.
  • To find out whether specially adapted information systems are used for the transport of dangerous goods.
  • To determine whether accidents with dangerous goods have occurred in the railway undertaking in the last year, as well as the common causes of their accidents.
  • Find out if additional training is provided for train drivers to transport dangerous goods.
  • To find out which factors of transport of dangerous goods by rail have the greatest impact on the accident.”

Point 7. Research results: there should be a connection between the research objectives (research question) you state in the first paragraph, and the results of your study. This connection is rather weak in your manuscript.

Response: Thank you for this comment. It was done. As I supplemented the objectives of the study, the results of the study are now related to the Research methodology.

Point 8. Please rename "limitation and recommendation" to "practical implications". You need to make sure the points mentioned in this part are well supported by the results in the previous section.

Response: Thank you for this helpful comment. It was done.

Point 9. In the last two sections, you should not use bullet points and numbering. I could spot some minor plagiarism in some sections of your work. Please make sure you are paraphrasing the sentences if they are taken from other studies.

Response: Thank you for this comment. There is no plagiarism here, it has already been checked. All sentences were paraphrased. The first version of the manuscript contained numbers that, in the reviewer’s opinion, needed to be changed to bullet points.

Point 10. Please make sure a decent language-proof reading service is used before your next submission.

Response: Thank you for this comment. It was done.

I would like to thank you for your constructive help once again.

Reviewer 2 Report

Author improved the paper according to reviewer comments. So, in my opinion, paper can be published in the present form. My only one remark is dedicated to the numbers in the Conclusion chapter. I would suggest replacing them with bullet items (the numbers in the paper are reserved to mark chapters and subsections).

Author Response

Response to Reviewer 2 Comments

Thank you very much for your valuable comments on the manuscript (843293) with the title “Improving Safety of Transportation of Dangerous Goods by Railway Transport”. I have tried my best to revise and improve the manuscript and made many changes in the manuscript according to your helpful comments.

Author improved the paper according to reviewer comments. So, in my opinion, paper can be published in the present form.

My only one remark is dedicated to the numbers in the Conclusion chapter. I would suggest replacing them with bullet items (the numbers in the paper are reserved to mark chapters and subsections).

Response: Thank you very much for your positive comments. I replaced the numbers with bullet items in the Conclusion chapter.

I would like to thank you for your constructive help once again.

Reviewer 3 Report

I would like to thank the author for addressing most of my comments. The manuscript is much improved in the revised version. 

Point 2. You use the term “Dangerous Freights” and not “Dangerous Goods” in your manuscript although you refer to RID regulation. Please provide the reason why you did that. My recommendation it to use the term “Dangerous Goods”. Please note that you already use the term “goods” (please see page 2, lines 46 and 47, lines 69 and 70).

Response from the author:

Thank you for this constructive suggestion. I changed the term “Freights” everywhere to the term “Goods”. At the same time, the title of the manuscript changed: Improving Safety of Transportation of Dangerous Goods by Railway Transport.

My new comment:

The term “Dangerous Freights” still appears in many parts of the manuscript despite the effort made by the author to use only the term “Dangerous Goods” (e.g, Page 1, line 20 - Page 1, line 35  - Page 5, line 211 - Page 9, line 342 ).

Point 6. Page 2, line 77: The title of Section 2 is “Literature review” scientific literature but in many cases in your text you use the term “scientific literature” (e.g., page 2, lines 75 and 90). Please retain “literature”.

Response from the author:

Thank you for this comment. I changed as you advised.

My new comment:

The term “scientific literature” still appears in the manuscript despite the effort made by the author to use only the term “literature review” (please see Abstract, line 13).

Point 17. Page 6, Table 1, First row: Words separation is incorrect.

Response from the author:

Thank you for this comment. I changed and arranged the words in first row of Table 1.

My new comment:

There is still a problem in Table 1. Evaluations factors by experts’ (please see columns 7 and 8).

Point 18. Page 7, Figure 4: There is no need for Figure 4 since the respective information is presented in Table 1.

Response from the author:

I think the main factors should be shown graphically as it would be more understandable and clearer to the reader.

My new comment:

Please reconsider your decision to keep Figure 4. I am not convinced that the specific Figure offers something different to the reader compared to Table 1.

Finally, the response letter to the reviewers should also include the new line numbers and not just the text. 

Author Response

Response to Reviewer 3 Comments

Thank you very much for your valuable comments on the manuscript (843293) with the title “Improving Safety of Transportation of Dangerous Goods by Railway Transport”. I have tried my best to revise and improve the manuscript and made many changes in the manuscript according to your helpful comments.

Point 1. The term “Dangerous Freights” still appears in many parts of the manuscript despite the effort made by the author to use only the term “Dangerous Goods” (e.g, Page 1, line 20 - Page 1, line 35  - Page 5, line 211 - Page 9, line 342 ).

Response: Thank you. Now all terms from “Dangerous freights” have been changed to “Dangerous goods” (Page 1, line 20, Page 1, line 35, Page 5, line 211, Page 9, line 342). In a new version of Manuscript - Page 1, line 18, Page 1, line 41, Page 6, line 228, Page 9, line 368.

Point 2. The term “scientific literature” still appears in the manuscript despite the effort made by the author to use only the term “literature review” (please see Abstract, line 13).

Response: Thanks, I also changed these words in the Abstract (Line 13).

Point 3. There is still a problem in Table 1. Evaluations factors by experts’ (please see columns 7 and 8).

Response: Thank you for this comment. The text in columns 7 and 8 has been corrected (line 312).

Point 4.  Please reconsider your decision to keep Figure 4. I am not convinced that the specific Figure offers something different to the reader compared to Table 1.

Response: Thank you. Figure 4 is now deleted (lines 290-292).

Point 5. Finally, the response letter to the reviewers should also include the new line numbers and not just the text.

Response: Thank you for this comment. I tried to do that.

I would like to thank you for your constructive help once again.

Round 2

Reviewer 1 Report

Except for some minor problems, the other comments are addressed.

You need to provide the relevant citations to the information you provided in the first two paragraphs. For example the following sentences: 1. "Rail transport is superior to other modes of transport because it is universal, reliable, able to transport dangerous goods regardless of the time of year, climatic conditions and transport options" 2. "Due to the dangerous nature of goods, dangerous accidents inland transport often have dire consequences for the population and the environment"

Point 4 is not addressed effectively. You need to cite the most relevant works, and then conclude the same way you have done in the last paragraph of the literature "The articles mainly analyze the requirements for land transport in general, but do not single out the main risks and the main factors that determine the likelihood of dangerous goods accidents by rail, nor do they provide recommendations on how to ensure the safety of dangerous goods by rail."

You are not improving the safety of hazardous material transportation by the railway. Instead, you are providing a technical note to the subject matter. Therefore, I recommend you to reconsider the title.

Once again, it is not recommended to use bullet points and numbering in your manuscript, as you did in the practical implications and conclusion sections. Last comment: plagiarism has to be double-checked again!

Author Response

Response to Reviewer Comments

Thank you very much for your valuable comments on the manuscript (843293) with the title “Improving Safety of Transportation of Dangerous Goods by Railway Transport”. I have tried my best to revise and improve the manuscript and made many changes in the manuscript according to your helpful comments.

Point 1. You need to provide the relevant citations to the information you provided in the first two paragraphs. For example, the following sentences: 1. "Rail transport is superior to other modes of transport because it is universal, reliable, able to transport dangerous goods regardless of the time of year, climatic conditions and transport options" 2. "Due to the dangerous nature of goods, dangerous accidents inland transport often have dire consequences for the population and the environment"

Response: Thank you for this comment. I cannot cite these sentences (lines 29-30 and lines 34-36), because they are not taken from scientific articles. This is taken from my lectures.

Point 2. Point 4 is not addressed effectively. You need to cite the most relevant works, and then conclude the same way you have done in the last paragraph of the literature "The articles mainly analyze the requirements for land transport in general, but do not single out the main risks and the main factors that determine the likelihood of dangerous goods accidents by rail, nor do they provide recommendations on how to ensure the safety of dangerous goods by rail."

Response: Thank you for this helpful comment. I corrected this part and tried to strengthen the academic gap as follows:

“Researchers emphasize that special attention needs to be paid to the transport of dangerous goods not only during the preparation and loading of such cargo, but also during transportation until the cargo reaches its final destination [19-24]. Blanco [25] argues that loading and unloading dangerous goods is less dangerous than transportation by road, but also covering some of the potential risks posed by these operations. However, the author does not examine more factors related to the safety of the transport of dangerous goods. In addition, it analyzes only the road safety aspects.” (Lines 138-143).

“Zhao et al. [32] found that the three most influential factors in Hazardous materials transportation accidents were human factors, the transport vehicle and facilities, and packing and loading of the Hazardous materials in China, but these factors are more related to road transport.” (Lines 153-155).

Point 3.  You are not improving the safety of hazardous material transportation by the railway. Instead, you are providing a technical note to the subject matter. Therefore, I recommend you to reconsider the title.

Response: Based on the results of the research, the main factors that have the greatest impact on the risk and occurrence of the accident were identified. As a result, recommendations have been made on how to transport dangerous goods safer by rail. I think there may be such a title.

Point 4.  Once again, it is not recommended to use bullet points and numbering in your manuscript, as you did in the practical implications and conclusion sections. Last comment: plagiarism has to be double-checked again!

Response: Thank you for this comment. All sentences were paraphrased. The bullet points are no longer used in the practical implications and conclusion sections.  

I would like to thank you for your constructive help once again.

This manuscript is a resubmission of an earlier submission. The following is a list of the peer review reports and author responses from that submission.

Round 1

Reviewer 1 Report

The abstract part should be shortened and restructured. You may start with a background sentence and the importance of the study; then state your research objective and contribution. The method you applied should come next following by the MAJOR findings of your research.

You may add "Hazardous Material" as a keyword.

Introduction: should be restructured as well. The classes of hazardous material should be moved into the background sub-section in the literature review section.

You need to follow a story-telling manner to attract the readers to read the rest of the manuscript.

In the first paragraph, you need to start with a more general background to the transportation of hazardous material, and then provide evidence that the railway transportation for such goods is preferred to the other modes of transportation.

One important aspect that needs to be considered in the paper is the risk of population exposure. I recommend the author to refer to "Pourhejazy, P., Kwon, O. K., & Lim, H. (2019). Integrating sustainability into the optimization of fuel logistics networks. KSCE Journal of Civil Engineering23(3), 1369-1383." and the references wherein. This aspect can be mentioned at the end of line 54, citing the relevant paper.

After stating the problem, and the importance of the study, one needs to review the most relevant research works in the literature with a critical lens. This will help the readers to understand what is missing, which you are going to address in your study. On this basis, you will have to state your contribution.

You may close the introduction part providing a paragraph to summarize the outline of this manuscript.

Logical order should be followed to re-write the literature review part. You may start with a background to the problem, including the classes of hazardous material, and then categorize the existing research works, review them in a structured manner. In the current state, I cannot see any LOGIC in the way you reviewed the papers. Besides, I feel that more studies should be reviewed in this section. Following

The contents of Section 3 should be moved into the introduction and methodology sections. Including a separate "Object and problem analysis" does not sounds good.

The methodology section should be more informative. You should consider that your potential readers may not be familiar with the method. You also need to elaborate on the methods you used for data processing. What you provided is only about data collection. There should be some theories involved in your research method. They need to be explained here.

It is unclear if the contents of Section 6 are the "practical implications" of your study. If yes, why are you providing content from earlier research works? 

The conclusion section needs to be improved. First of all, the findings you mentioned here should be well supported by your numerical analysis. Second, you need to provide the limitations of your work. On this basis, you are recommended to provide insight for future research works. 

General comment: Overall, punctuation should be improved; the article "the" is missing in many cases throughout the paper. I could also spot many typos and grammar mistakes; language proof-reading service could certainly help.

Reviewer 2 Report

Interesting and well-constructed paper. The main aim of the reviewed paper was to analyze and identify the main risks associated with the transport of dangerous goods by rail, as well as to identify and assess the main factors of safe transport in order to reduce the risk of accidents and provide recommendations on how to transport dangerous goods
safely. In my opinion, paper can be published after taking into account the following remarks:

  • on the figures in the whole paper Author should complete the names of the axes "x" and "y" and units,
  • I don't know why some part of the paper text is in red colour ?- it should be converted on black colour,

- the Author didn't refer in the literature review of the subject for the transport of dangerous materials on first and last mile delivery.
In chapter 2 "Literature review", a short paragraph should be added at the end of the chapter regarding the transport of dangerous materials on first and last mile delivery. It is enough to add that ensuring the transport of dangerous materials is important at every stage of the supply chain, including first and last mile delivery, f.ex .:

  • Slabinac M.: Innovative solutions for a  "Last-mile" delivery a European experience. Business Logistics in Modern Management. Proceedins of 15th international scientific conference Business Logistics in Modern Management, p. 111-129.
  • Macioszek E.: First and last mile delivery - problems and issues. [in:] G. SierpiÅ„ski (ed.) Advanced Solutions of Transport Systems for Growing Mobility. Advances in Intelligent Systems and Computing 631. Springer International Publishing Switzerland 2018, p. 147-154.
  • Schliwa G., Armitage R., Aziz S., Evans J., Rhoades J.: Sustainable city logistics - making cargo cycles viable for urban freight transport. Research in Transportation Busines & Management, vol. 15, p. 50-57, 2015.
  •  
  • the Author didn't explain why the number of experts was 47? On what basis was the number of experts selected?

Reviewer 3 Report

Page 1 Introduction: You have to say a few things about the “International Carriage of Dangerous Goods by Rail (RID)” Regulation.

You use the term “Dangerous Freights” and not “Dangerous Goods” in your manuscript although you refer to RID regulation. Please provide the reason why you did that. My recommendation it to use the term “Dangerous Goods”. Please note that you already use the term “goods” (please see page 2, lines  46 and 47, lines 69 and 70).

Page 2, line 44: “Transportation of dangerous freights by rail is a process that requires safety.”. This sentence is too general. The basic point here is that “transportation of dangerous goods” is characterized by “special safety requirements”. You must emphasize that from the beginning.

Page 3, line 105: “……until the cargo reaches its final destination [10, 18-25].” Please provide some more text concerning the 9 references included in the brackets.  The addition of a sentence or two for each reference will improve “Section 2. Literature review”.

Please check your manuscript throughout for English grammar and syntax errors (e.g., “evaluation of experts” or “evaluation by experts” ?) (e.g., page 2, line 84, “Diernhofer et al. [10] states…” must be ““Diernhofer et al. [10] state…). (Page 4, line 150, “In Figure 2 shows that the number…” must be “Figure 2 shows that the number…”). (Page 4, line 156, “An expert is called a specialist….” must be “We called someone as an expert when ….”).(Page 8, line 310, “The questionnaire found that the greatest risks were…”must be “According to the results of the questionnaire it was found …..” ).

Page 2, line 77: The title of Section 2 is “Literature review” scientific literature but in many cases in your text you use the term “scientific literature” (e.g., page 2, lines 75 and 90). Please retain “literature”.

Page 3, line 100: “The authors Ghazinoory and Kheirkhah [17] agree…” must be “Ghazinoory and Kheirkhah [17] agree…”.

Page 3, line 105: “Conca et al. [26] states…” must be “Conca et al. [26] state…”.

Page 3, line 107: “The author Najib et al. [27] suggests…” must be “Najib et al. [27] suggest…”

Page 3, line 112: “The authors Bouissou and co-authors [31] describe…” must be “Bouissou et al. [31] describe…”

Page 3, line 115: “Tomasoni et al. [32] describes…” must be “Tomasoni et al. [32] describe…”

Page 5, line 171: “The survey conducted in the autumn of 2019 involved 47 experts”. Please include the criteria for the selection of experts.

Page 7, line 247: “The author Blanco [41] in his article presents…” must be “Blanco [41] presents…”

General comment: Please check all the similar cases throughout your text.

Page 3, line 119: “Title of Section 3. Object and problem analysis”. What do you mean by “Object” ?

Page 3, Figure 1: Data presented in Figure 1 refer to period 2013-2017. Please add data for the years 2018 and 2019 if available.

Page 4, line 141: “Incidents and accidents”. Please define “incidents” so the reader can understand the difference with “accidents”.

Page 4, Figure 2: Since Figure 2 refers to rail only, please modify the title of Figure 2. The title refers to all transport modes as it is now.

Page 4: It seems that something went wrong with the bullet points presented at the bottom of page 4.

Page 5, Figure 3: The sum of all percentages presented in Figure 3 is 78%. Percentage for class 9 is missing.

Page 6, Table 1, First row: Words separation is incorrect.

Page 7, Figure 4: There is no need for Figure 4 since the respective information is presented in Table 1.

Page 8, lines 285, 286: “An important task is to assess the risk. This can be done in different ways and using different models and methods, depending on certain risk factors and the results to be achieved.”. Please be more specific and provide detailed information concerning “which models” and “which methods”.

Section 7. Discussion and Conclusions: Please include the limitations and constraints of your research.